# Provably Efficient Online Hyperparameter Optimization with Population-Based Bandits

**Jack Parker-Holder**
University of Oxford
jackph@robots.ox.ac.uk

**Vu Nguyen**
University of Oxford
vu@robots.ox.ac.uk

**Stephen J. Roberts**
University of Oxford
sjrob@robots.ox.ac.uk

## Abstract

Many of the recent triumphs in machine learning are dependent on well-tuned hyperparameters. This is particularly prominent in reinforcement learning (RL) where a small change in the configuration can lead to failure. Despite the importance of tuning hyperparameters, it remains expensive and is often done in a naive and laborious way. A recent solution to this problem is Population Based Training (PBT) which updates both weights and hyperparameters in a *single training run* of a population of agents. PBT has been shown to be particularly effective in RL, leading to widespread use in the field. However, PBT lacks theoretical guarantees since it relies on random heuristics to explore the hyperparameter space. This inefficiency means it typically requires vast computational resources, which is prohibitive for many small and medium sized labs. In this work, we introduce the first provably efficient PBT-style algorithm, Population-Based Bandits (PB2). PB2 uses a probabilistic model to guide the search in an efficient way, making it possible to discover high performing hyperparameter configurations with far fewer agents than typically required by PBT. We show in a series of RL experiments that PB2 is able to achieve high performance with a modest computational budget.

## 1 Introduction

Deep neural networks [22, 26, 38] have achieved remarkable success in a variety of fields. Some of the most notable results have come in reinforcement learning (RL), where the last decade has seen a series of significant achievements in games [60, 48, 8] and robotics [51, 33]. However, it is notoriously difficult to reproduce RL results, often requiring excessive trial-and-error to find the optimal hyperparameter configurations [6, 24, 50].

This has led to a surge in popularity for Automated Machine Learning (AutoML, [29]), which seeks to automate the training of machine learning models. A key component in AutoML is automatic hyperparameter selection [7, 47], where popular approaches include Bayesian Optimization (BO, [10, 25, 49]) and Evolutionary Algorithms (EAs, [12, 27]). Using automated methods for RL (AutoRL) is crucial for accessibility and for generalization, since different environments typically require totally different hyperparameters [24]. Furthermore, it may even be possible to improve performance of existing methods using learned parameters. In fact, BO was revealed to play a valuable role in AlphaGo, improving the win percentage before the final match with Lee Sedol [13].

A particularly promising approach, Population Based Training (PBT, [32, 39]), showed it is possible to achieve impressive performance by updating both weights and hyperparameters during a *single* training run of a population of agents. PBT works in a similar fashion to a human observing experiments, periodically replacing weaker performers with superior ones. PBT has shown to be particularly effective in reinforcement learning, and has been used in a series of recent works to improve performance [56, 44, 17, 31].

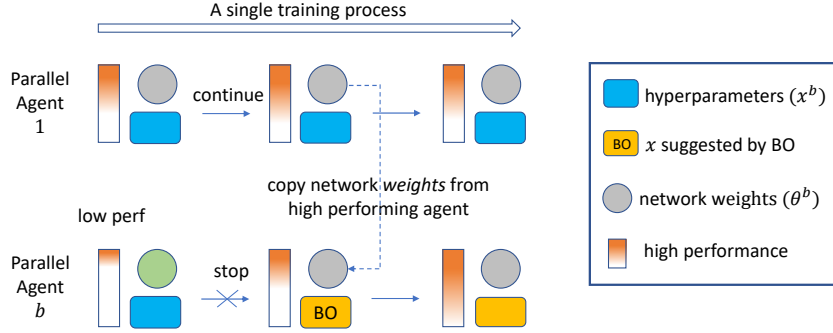

Figure 1: Population-Based Bandit Optimization: a population of agents is trained in parallel. Each agent has weights (grey) and hyperparameters (blue). The agents are evaluated periodically (orange bar), and if an agent is underperforming, it's weights are replaced by randomly copying one of the better performing agents, and its hyperparameters are selected using Bayesian Optimization.

However, PBT's Achilles heel comes from its reliance on heuristics for exploring the hyperparameter space. This essentially creates new meta-parameters, which need to be tuned. In many cases, PBT underperforms a random baseline without vast computational resources, since small populations can collapse to a suboptimal mode. In addition, PBT lacks theoretical grounding, given that greedy exploration methods suffer from unbounded regret.

Our key contribution is the first provably efficient PBT-style algorithm, Population-Based Bandit Optimization, or PB2 (Fig. 1). To do this, we draw the connection between maximizing the reward in a PBT-style training procedure to minimizing the regret in bandit optimization [61, 14]. Formally, we consider the online hyperparameter selection problem as batch Gaussian Process bandit optimization of a time-varying function. PB2 is more computationally efficient than the existing batch BO approaches [15, 20] since it can (1) learn the optimal schedule of hyperparameters and (2) optimize them in a single training run. We derive a bound on the cumulative regret for PB2, the first such result for a PBT-style algorithm. Furthermore, we show in a series of RL experiments that PB2 is able to achieve high rewards with a modest computational budget.

## 2 Problem Statement

In this paper, we consider the problem of selecting optimal hyperparameters, $x_t^b$, from a compact, convex subset $\mathcal{D} \in \mathbb{R}^d$ where $d$ is the number of hyperparameters. Here the index $b$ refers to the $b$th agent in a population/batch, and the subscript $t$ represents the number of timesteps/epochs/iterations elapsed during the training of a neural network. Particularly, we consider the *schedule* of optimal hyperparameters over time $\left(x_t^b\right)_{t=1,...T}$.

Population-Based Training (PBT, [32]) is a well-known algorithm for learning a schedule of hyperparameters by training a population (or batch) of $B$ agents in parallel. Each agent $b \in B$ has both hyperparameters $x_t^b \in \mathbb{R}^d$ and weights $\theta_t^b$. At every $t_{\text{ready}}$ step interval (i.e. if $t \mod t_{\text{ready}} = 0$), the agents are ranked and the worst performing agents are replaced with members of the best performing agents ($A \subset B$) as follows:

- **Weights** ($\theta_t^b$): copied from one of the best performing agents, i.e. $\theta_t^j \sim \text{Unif}\{\theta_t^j\}_{j \in A}$.
- **Hyperparameters** ($x_t^b$): with probability $\epsilon$ it uses **random exploration**, and re-samples from the original distribution, otherwise, it uses **greedy exploration**, and perturbs one of the best performing agents, i.e $\{x^j \times \lambda\}_{j \in A}, \lambda \sim [0.8, 1.2]$.

This leads to the learning of hyperparameter schedules – a single agent can have different configurations at different time-steps during a single training process. This is important in the context of training deep reinforcement learning agents as dynamic hyperparameter schedules have been shown to be effective [52, 32, 17]. On the other hand, most of the existing hyperparameter optimization approaches aim to find a fixed set of hyperparameters over the course of optimization.

To formalize this problem, let $F_t(x_t)$ be an objective function under a given set of hyperparameters at timestep $t$. An example of $F_t(x_t)$ could be the reward for a deep RL agent. When training for a

total of $T$ steps, our goal is to maximize the final performance $F_T(x_T)$. We formulate this problem as optimizing the time-varying black-box reward function $f_t$, over $\mathcal{D}$. Every $t_{\text{ready}}$ steps, we observe and record noisy observations, $y_t = f_t(x_t) + \epsilon_t$, where $\epsilon_t \sim \mathcal{N}(0, \sigma^2 \mathbf{I})$ for some fixed $\sigma^2$. The function $f_t$ represents the change in $F_t$ after training for $t_{\text{ready}}$ steps, i.e. $F_t - F_{t-t_{\text{ready}}}$. We define the best choice at each timestep as $x_t^* = \arg \max_{x_t \in \mathcal{D}} f_t(x_t)$, and so the **regret** of each decision as $r_t = f_t(x_t^*) - f_t(x_t)$.

**Lemma 1.** *Maximizing the final performance $F_T$ of a model with respect to a given hyperparameter schedule $\{x_t\}_{t=1}^T$ is equivalent to maximizing the time-varying black-box function $f_t(x_t)$ and minimizing the corresponding cumulative regret $r_t(x_t)$,*

$$\max F_T(x_T) = \max \sum_{t=1}^T f_t(x_t) = \min \sum_{t=1}^T r_t(x_t). \tag{1}$$

In subsequent sections, we present a time-varying bandit approach which is used to minimize the cumulative regret $R_T = \sum_{t=1}^T r_t$. Lemma 1 shows this is equivalent to maximizing the final performance/reward of a neural network model (see: Section 9 in the appendix for the proof).

## 3   Population-Based Bandit Optimization

We now introduce Population-Based Bandit Optimization (PB2) for optimizing the hyperparameter schedule $\left(x_t^b\right), \forall t = 1, ..., T$ using parallel agents $b = 1, ..., B$. After each agent $b$ completes $t_{\text{ready}}$ training steps, we store the data $(y_t^b, t, x_t^b)$ in a dataset $D_t$ which will be used to make an informed decision for the next set of hyperparameters.

We below present the mechanism to select the next hyperparameters for parallel agents. Motivated by the equivalence of the maximized reward and minimized cumulative regret in Lemma 1, we propose the parallel time-varying bandit optimization.

### 3.1   Parallel Gaussian Process Bandits for a Time-Varying Function

We first describe the time-varying Gaussian process as the surrogate models, then we extend it to the parallel setting for our PB2 algorithm.

**Time-varying Gaussian process as the surrogate model.**   Following previous works in the GP-bandit literature [61], we model $f_t$ using a Gaussian Process (GP, [54]) which is specified by a mean function $\mu_t : \mathcal{X} \to \mathbb{R}$ and a kernel (covariance function) $k : \mathcal{D} \times \mathcal{D} \to \mathbb{R}$. If $f_t \sim GP(\mu_t, k)$, then $f_t(x_t)$ is distributed normally $\mathcal{N}(\mu_t(x_t), k(x_t, x_t))$ for all $x_t \in \mathcal{D}$. After we have observed $T$ data points $\{(x_t, f(x_t))\}_{t=1}^T$, the GP posterior belief at new point $x_t' \in \mathcal{D}$, $f_t(x_t')$ follows a Gaussian distribution with mean $\mu_t(x')$ and variance $\sigma_t^2(x')$ as:

$$\mu_t(x') := \mathbf{k}_t(x')^T (\mathbf{K}_t + \sigma^2 \mathbf{I})^{-1} \mathbf{y}_t \tag{2}$$

$$\sigma_t^2(x') := k(x', x') - \mathbf{k}_t(x')^T (\mathbf{K}_t + \sigma^2 \mathbf{I})^{-1} \mathbf{k}_t(x'), \tag{3}$$

where $\mathbf{K}_t := \{k(x_i, x_j)\}_{i,j=1}^t$ and $\mathbf{k}_t := \{k(x_i, x_t')\}_{i=1}^t$. The GP predictive mean and variance above will later be used to represent the exploration-exploitation trade-off in making decision under the presence of uncertainty.

To represent the non-stationary nature of a neural network training process, we cast the problem of optimizing neural network hyperparameters as time-varying bandit optimization. We follow [9] to formulate this problem by modeling the reward function under the time-varying setting as follows:

$$f_1(x) = g_1(x), \quad f_{t+1}(x) = \sqrt{1 - \omega} f_t(x) + \sqrt{\omega} f g_{t+1}(x) \quad \forall t \geq 2, \tag{4}$$

where $g_1, g_2, ...$ are independent random functions with $g \sim GP(0, k)$ and $\omega \in [0, 1]$ models how the function varies with time, such that if $\omega = 0$ we return to GP-UCB and if $\omega = 1$ then each evaluation is completely independent. This model introduces a new hyperparameter ($\omega$), however, we note it can be optimized by maximizing the marginal likelihood for a trivial additional cost compared to the expensive function evaluations (we include additional details on this in the Appendix). This leads to the extensions of Eqs. (2) and (3) using the new covariance matrix $\tilde{\mathbf{K}}_t = \mathbf{K}_t \circ \mathbf{K}_t^{\text{time}}$ where $\mathbf{K}_t^{\text{time}} = [(1 - \omega)^{|i-j|/2}]_{i,j=1}^T$ and $\tilde{\mathbf{k}}_t(x) = \mathbf{k}_t \circ \mathbf{k}_t^{time}$ with $\mathbf{k}_t^{time} = [(1 - \omega)^{(T+1-i)/2}]_{i=1}^T$. Here $\circ$ refers to the Hadamard product.

**Selecting hyperparameters for parallel agents.** In the PBT setting, we consider an entire population of parallel agents. This changes the problem from a sequential to a *batch* blackbox optimization. This poses an additional challenge to select multiple points simultaneously $x_t^b$ without full knowledge of all $\{(x_t^j, y_t^j)\}_{j=1}^{j-1}$. A key observation in [14] is that since a GP's variance (Eqn. 3) does not depend on $y_t$, the acquisition function can account for incomplete trials by updating the uncertainty at the pending evaluation points. Concretely, we define $x_t^b$ to be the $b$-th point selected in a batch, after $t$ timesteps. This point may draw on information from $t + (b - 1)$ previously selected points. In the single agent, sequential case, we set $B = 1$ and recover $t, b = t - 1$. Thus, at the iteration $t$, we find a next batch of $B$ samples $\left[x_t^1, x_t^2, ... x_t^B\right]$ by sequentially maximizing the following acquisition function:

$$x_t^b = \underset{x \in \mathcal{D}}{\arg\max}\, \mu_{t,1}(x) + \sqrt{\beta_t}\sigma_{t,b}(x), \forall b = 1, ...B \tag{5}$$

for $\beta_t > 0$. In Eqn. (5) we have the mean from the previous batch ($\mu_{t,1}(x)$) which is fixed, but can update the uncertainty using our knowledge of the agents currently training ($\sigma_{t,b}(x)$). This significantly reduces redundancy, as the model is able to explore distinct regions of the space.

### 3.2 PB2 Algorithm and Convergence Guarantee

To estimate the GP predictive distribution in Eqs. (2, 3), we use the product form $\tilde{k} = k^{\mathrm{SE}} \circ k^{\mathrm{time}}$ [35, 9], which considers the time varying nature of training a neural network. Then, we use Eqn. (5) to select a batch of points by utilizing the reduction in uncertainty for models currently training. As far as we know, this is the first use of a time-varying kernel in the PBT-style setting. Unlike PBT, this allows us to efficiently make use of data from previous trials when selecting new configurations, rather than reverting to a uniform prior, or perturbing existing configurations. The full algorithm is shown in Algorithm 1.

---

**Algorithm 1:** Population-Based Bandit Optimization (PB2)

**Initialize:** Network weights $\{\theta_0^b\}_{b=1}^B$, hyperparameters $\{x_0^b\}_{b=1}^B$, dataset $D_0 = \emptyset$
**(in parallel) for** $t = 1, \ldots, T - 1$ **do**

 1. **Update Models:** $\theta_t^b \leftarrow \mathrm{step}(\theta_{t-1}^b | x_{t-1}^b)$
 2. **Evaluate Models:** $y_t^b = F_t(x_t^b) - F_{t-1}(x_{t-1}^b) + \epsilon_t$ for all $b$
 3. **Record Data:** $D_t = D_{t-1} \cup \{(y_t^b, t, x_t^b)\}_{b=1}^B$
 4. If $t \mod t_{\mathrm{ready}} = 0$:

 - **Copy weights:** Rank agents, if $\theta^b$ is in the bottom $\lambda\%$ then copy weights $\theta^j$ from the top $\lambda\%$.

 - **Select hyperparameters:** Fit a GP model to $D_t$ and select hyper-parameters $x_t^b, \forall b \leq B$ by maximizing Eq. (5).

**Return the best trained model** $\theta$

---

Next we present our main theoretical result, showing that PB2 is able to achieve sublinear regret when the time-varying function is correlated. This is the first such result for a PBT-style algorithm.

**Theorem 2.** *Let the domain* $\mathcal{D} \subset [0, r]^d$ *be compact and convex where* $d$ *is the dimension and suppose that the kernel is such that* $f_t \sim GP(0, k)$ *is almost surely continuously differentiable and satisfies Lipschitz assumptions* $\forall L \geq 0, t \leq \mathcal{T}, p(\sup \left| \frac{\partial f_t(\boldsymbol{\beta})}{\partial \boldsymbol{\beta}^{(k)}} \right| \geq L_t) \leq ae^{-(L_t/b)^2}$ *for some* $a, b$. *Pick* $\delta \in (0, 1)$, *set* $\beta_T = 2 \log \frac{\pi^2 T^2}{2\delta} + 2d \log rdbT^2 \sqrt{\log \frac{da\pi^2 T^2}{2\delta}}$ *and define* $C_1 = 32/\log(1 + \sigma_f^2)$, *the PB2 algorithm satisfies the following regret bound after* $T$ *time steps over* $B$ *parallel agents with probability at least* $1 - \delta$:

$$R_{TB} = \sum_{t=1}^T f_t(\mathbf{x}_t^*) - f_t(\mathbf{x}_t) \leq \sqrt{C_1 T \beta_T \left( \frac{T}{\tilde{N}B} + 1 \right) \left( \gamma_{\tilde{N}B} + \left[ \tilde{N}B \right]^3 \omega \right)} + 2$$

*the bound holds for any block length* $\tilde{N} \in \{1, ..., T\}$ *and* $B \ll T$.

This result shows the regret for PB2 decreases as we increase the population size $B$. This should be expected, since adding computational resources should benefit final performance (which is equivalent to minimizing regret). We demonstrate this property in Table 1. When using single agent $B = 1$, our bound becomes the time-varying GP-UCB setting in [9].

In our setting, if the time-varying function is highly correlated, i.e., the information between $f_1(.)$ and $f_T(.)$ does not change significantly, we have $\omega \to 0$ and $\tilde{N} \to T$. Then, the regret bound grows sublinearly with the number of iterations $T$, i.e., $\lim_{T\to\infty} \frac{R_{TB}}{TB} = 0$. On the other hand (in the worst case), if the time-varying function is not correlated at all, such as $\tilde{N} \to 1$ and $\omega \to 1$, then PB2 achieves linear regret [9].

**Remark.** This theoretical result is novel and significant in two folds. First, by showing the equivalent between maximizing reward and minimizing the bandit regret, this regret bound quantifies the gap between the optimal reward and the achieved reward (using parameters selected by PB2). The regret bound extends the result established by [61, 9], to a more general case with parallelization. To the best of our knowledge, this is the first kind of convergence guarantee in a PBT-style algorithm.

Our approach is advantageous against all existing batch BO approaches [14, 20] in that PB2 considers optimization in a *single training run* of parallel agents while the existing works need to evaluate using *multiple training runs* of parallel agents which are more expensive. In addition, PB2 can learn a *schedule* of hyperparameters while the existing batch BO can only learn a static configuration.

## 4   Related Work

**Hyperparameter Optimization.** Hyperparameter optimization [34, 7, 30] is a crucial component of any high performing machine learning model. In the past, methods such as grid search and random search [5] have proved popular, however, with increased focus on Automated Machine Learning (AutoML, [29]), there has been a great deal of progress moving beyond these approaches. This success has led to a variety of tools becoming available [19, 45].

**Population Based Approaches.** Taking inspiration from biology, population-based methods have proved effective for blackbox optimization problems [46]. Evolutionary Algorithms (EAs, [3]) take many forms, one such method is Lamarckian EAs [65] in which parameters are inherited whilst hyperparameters are evolved. Meanwhile, other methods learn both hyperparameters and weights [12, 27]. These works motivate the recently introduced PBT algorithm [32], whereby network parameters are learned via gradient descent, while hyperparameters are evolved. Its key strengths lie in the ability to learn high performing hyperparameter schedules in a single training run, leading to strong performance in a variety of settings [44, 39, 17, 56]. PBT works by focusing on high performing configurations, however, this greedy property means it is unlikely to explore areas of the search space which are "late bloomers". In addition, the core components of the algorithm rely on handcrafted meta-hyperparameters, such as the degree of mutation. These design choices mean the algorithm lacks theoretical guarantees. We address these issues in our work.

**Bayesian Optimization.** Bayesian Optimization (BO [10, 59, 55]) is a sequential model-based blackbox optimization method [28]. BO works by building a surrogate model of the blackbox function, typically taken to be a Gaussian Process [54]. BO has been shown to produce state-of-the-art results in terms of sample efficiency, making dramatic gains in several prominent use cases [13]. Over the past few years there has been increasing focus on distributed implementations [21, 2] which seek to select a batch of configurations for concurrent evaluation. Despite this increased efficiency, these methods still require multiple training runs. Another recent method, Freeze-Thaw Bayesian Optimization [63] considers a 'bag' of current solutions, with their future loss assumed to follow an exponential decay. The next model to optimize is chosen via entropy maximisation. This approach bears similarities in principle with PBT, but from a Bayesian perspective. The method, however, makes assumptions about the shape of the loss curve, does not adapt hyperparameters during optimization (as in PBT) and is sequential rather than parallelized. Our approach takes appealing properties of both these methods, using the Bayesian principles but adapting in an online fashion.

**Hybrid Algorithms.** Bayesian and Evolutionary approaches have been combined in the past. In [53], the authors demonstrate using an early version of BO improves a simple Genetic Algorithm, while [1] shows promising results through combining BO and random search. In addition, the recently popular Hyperband algorithm [40], was shown to exhibit stronger performance with a BO controller [18]. The main weakness of these methods is their inability to learn schedules.

**Hyperparameter Optimization for Reinforcement Learning (AutoRL).** Finally, methods have been proposed specifically for RL, dating back to the early 1990s [62]. These methods have typically

had a narrower focus, optimizing an individual parameter. More recent work [52] proposes to adapt hyperparameters online, by exploring different configurations in an off-policy manner in between iterations. Additionally [16] concurrently proposed a similar to PBT, using evolutionary hyperparameter updates specifically for RL. These methods show the benefit of re-using samples, and we believe it would be interesting future work to consider augmenting our method with off-policy or synthetic samples (from a learned dynamics model) for the specific RL use case.

## 5  Experiments

We focus our experiments on the RL setting, since it is notoriously sensitive to hyperparameters [24]. We evaluate population sizes of $B \in \{4, 8\}$, which means the algorithm can be run locally on most modern computers. This is significantly less than the $B > 20$ used in the original PBT paper [32].

All experiments were conducted using the tune library [43, 42][1]. Our GP implementation is extended from GPy [23], where we use the squared exponential kernel in combination with the time kernel as described in Section 3. We optimize all GP-kernel hyperparameters, as well as the block length $\tilde{N}$, by maximizing the marginal likelihood [54].

### 5.1  On Policy Reinforcement Learning

We consider optimizing a policy for continuous control problems from the OpenAI Gym [11]. In particular, we seek to optimize the hyperparameters for Proximal Policy Optimization (PPO, [58]), for the following tasks: BipedalWalker, LunarLanderContinuous, Hopper and InvertedDoublePendulum.

Our primary benchmark is PBT, where we use an identical configuration to PB2 aside from the selection of $x_t^b$ (the explore step). We also compare against a random search (RS) baseline [5]. Random search is a challenging baseline because it does not make assumptions about the underlying problem, and typically achieves close to optimal performance asymptotically [29]. We include a Bayesian Optimization (BO) baseline which uses the Expected Improvement acquisition function. Finally, we also compare our results against a recent state-of-the-art distributed algorithm (ASHA, [41]). ASHA is one of the most recent distributed methods, shown to outperform PBT for supervised learning. We believe we are the first to test it for RL.

Table 1: Median best performing agent across 10 seeds. The best performing algorithms are bolded.

|  | $B$ | RS | BO | ASHA | PBT | PB2 | vs. PBT |
|---|---|---|---|---|---|---|---|
| BipedalWalker | 4 | 234 | 133 | 236 | 223 | **276** | +24% |
| LunarLanderContinuous | 4 | 161 | 206 | 213 | 159 | **235** | +48% |
| Hopper | 4 | 1638 | 1760 | 1819 | 1492 | **2346** | +57% |
| InvertedDoublePendulum | 4 | 8094 | 8607 | 7899 | **8893** | 8179 | -8% |
| BipedalWalker | 8 | 240 | 237 | 255 | 277 | **291** | +5% |
| LunarLanderContinuous | 8 | 175 | 240 | 231 | 247 | **275** | +11% |

For all environments we use a neural network policy with two 32-unit hidden layers and tanh activations. During training, we optimize the following hyperparameters: batch size, learning rate, GAE parameter ($\lambda$, [57]) and PPO clip parameter ($\epsilon$). We use the same fixed ranges across all four environments (included in the Appendix Section 8). All experiments are conducted for $10^6$ environment timesteps, with the $t_{\text{ready}}$ command triggered every $5 \times 10^4$ timesteps. For BO, we train each agent sequentially for $500k$ steps, and selects the best to train for the remaining budget. For ASHA, we initialize a population of 18 agents to compare against $B = 4$ and 48 agents for $B = 8$. These were chosen to achieve the same total budget with the grace period equal to the $t_{\text{ready}}$ criteria for PBT and PB2. Given the typically noisy evaluation of policy gradient algorithms [24] we repeat each experiment with ten seeds. We show the median best reward achieved from each run in Table 1 and plot the median best performing agent from each run, with the interquartile ranges (IQRs) in Fig. 2.

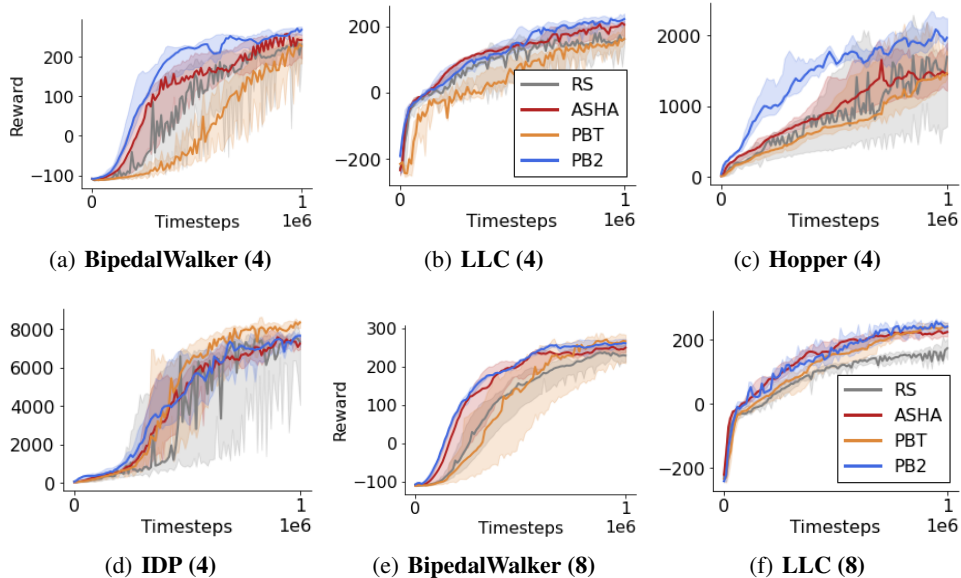

Figure 2: Median best performing agent across ten seeds, with IQR shaded. Population size ($B$) is shown in brackets.

In almost all cases we see performance gains from PB2 vs. PBT. In fact, $3/4$ of cases PBT actually underperforms random search with the smaller population size ($B = 4$), demonstrating its reliance on large computational resources. This is confirmed in the original PBT paper, where the smallest population size fails to outperform (see Table 1, [32]). One possible explanation for this is the greediness of PBT leads to prematurely abandoning promising regions of the search space. Another is that the small changes in parameters (multiple of $0.8$ or $1.2$) is mis-specified for discovering the optimal regions, thus requiring more initial samples to sufficiently span the space. This may also be the case if there is a shift later in the optimization process. Interestingly, PBT does perform well for InvertedDoublePendulum, this may be explained by the relative simplicity of the problem. We see that BO also performs well here, confirming that it may not require the same degree of adaptation during training as the other tasks (such as BipedalWalker).

For the larger population size we confirm the effectiveness of PBT, which outperforms ASHA and Random Search. However PB2 outperforms PBT by $+5\%$ and $+11\%$ for the BipedalWalker and LunarLanderContinuous environments respectively. This shows that PB2 is able to scale to larger computational budgets.

Interestingly, the state-of-the-art supervised learning performance of ASHA fails to translate to RL, where it clearly performs worse than both PBT and PB2 for the larger setting, and performs worse than PB2 for the smaller one.

**Robustness to Hyperparameter Ranges** One key weakness of PBT is a reliance on a large population size to explore the hyperparameter space. This problem can be magnified if the hyperparameter range is mis-specified or unknown (in a sense, the bounds placed on the hyperparameters may require tuning). PB2 avoids this issue, since it is able to select a point anywhere in the range, so does not rely on random sampling or gradual movements to get to optimal regions. We evaluate this by re-running the BipedalWalker task with a batch size drawn from $\{5000, 200000\}$. This means many agents are initialized in a very inefficient way, since when an agent has a batch size of $200,000$ it is using $20\%$ of total training samples for a single gradient step.

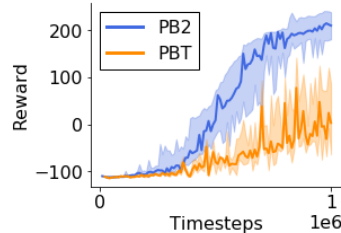

Figure 3: Median curves, IQR shaded.

As we see in Fig. 3 the performance for PBT is significantly reduced, while PB2 is still able to learn good policies. While both methods perform worse than in Table 1, PB2 still achieves a median best of 203, vs. $-12$ for PBT. This is a critical issue, since for new problems we will not know the optimal hyperparameter ranges ex-ante, and thus require a method which is capable of learning without this knowledge.

**Scaling to Larger Populations.** For PB2 to be broadly useful, it needs the ability to scale when more resources are available. To test this we repeated the BipedalWalker experiment with $B = 16$. As we see in Figure 4, both PBT and PB2 achieve optimal rewards ($> 300$), but PB2 is still more efficient. This is a promising initial result, although of course it will be interesting to test even larger settings in the future.

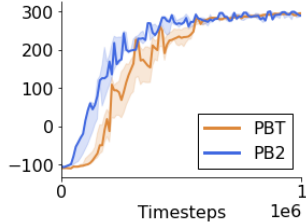

Figure 4: Median curves, IQR shaded.

## 5.2 Off Policy Reinforcement Learning

We now evaluate PB2 in a larger setting, optimizing hyperparameters for IMPALA [17]. in the breakout and SpaceInvaders environments from the Arcade Learning Environment [4]. In the original IMPALA paper, the best results come with the use of PBT with a population size of $B = 24$. Here we optimize the same three hyperparameters as in the original paper, but with a much smaller population ($B = 4$), making it essential to efficiently explore the hyperparameter space.

We train for 10 million timesteps, equivalent to 40 million frames, and set $t_{\text{ready}}$ to $5 \times 10^5$ timesteps. We repeat each experiment for 7 random seeds. PB2 achieves performance which is comparable with the hand-tuned performance reported in the rllib implementation,[2] while PBT underperforms, particularly in the SpaceInvaders environment.

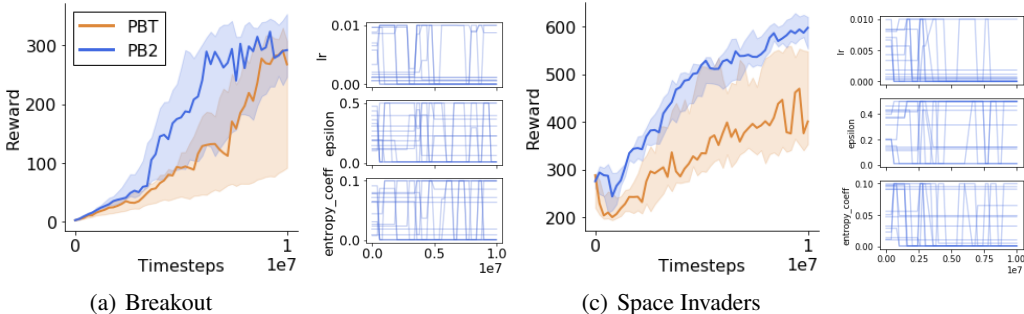

(a) Breakout       (c) Space Invaders

Figure 5: In both (a) and (b) we show training performance on the left, with median curves for seven seeds and inter-quartile range shaded. On the right, we show all agent configurations found by PB2.

For each environment in Fig. 5 we include the hyperparameters used for all agents across all seeds of training. PB2 effectively explores the entire range of parameters, which enables it to find optimal configurations even with a small number of trials. More details are in the Appendix, Section 8.

## 6 Conclusion

We introduced Population-Based Bandits (PB2), the first PBT-style algorithm with sublinear regret guarantees. PB2 replaces the heuristics from the original PBT algorithm with theoretically guided GP-bandit optimization. This allows it to balance exploration and exploitation in a principled manner, preventing mode collapse to suboptimal regions of the hyperparameter space and making it possible to find high performing configurations with a small computational budget. Our algorithm complements the existing approaches for optimizing hyperparameters for deep learning frameworks. We believe the gains for reinforcement learning will be particularly useful, given the number of hyperparameters present, and the difficulty in optimizing them with existing techniques.

Finally, we believe there are several future directions opened by taking our approach, such as updating population sizes based on the value of the acquisition function, and extending the search space to select the optimization algorithms or neural network architectures with BO. We also believe there may be further gains in reinforcement learning experiments through making use of off-policy data [52]. We leave these to exciting future work.

## Disclosure of Funding

Nothing to declare.

## Broader Impact

Population Based Training (PBT) has become a prominent algorithm in machine learning research, leading to gains in reinforcement learning (e.g. IMPALA [17]) and industrial applications (e.g. Waymo). As such, we believe the gains provided by PB2 will have a significant impact. We believe this work will allow labs with small to medium sized computational resources to gain the benefit of population-based training without the excessive computational cost required to ensure sufficient exploration. This should be particularly helpful for achieving competitive performance in reinforcement learning experiments. To aid this, our implementation is integrated with the widely used ray library [43].

## Footnotes

[1]See code here: https://github.com/jparkerholder/PB2

[2]See "RLlib IMPALA 32-workers" here: https://github.com/ray-project/rl-experiments

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
