[Supplementary Material]

# Appendix

## 7   Additional Experiments

**Supervised Learning** While the primary motivation for our work is RL, we also evaluated PB2 the supervised learning case, to test the generality of the method. Concretely, we used PB2 to optimize six hyperparameters for a Convolutional Neural Network (CNN) on the CIFAR-10 dataset [37]. In each setting we randomly sample the initial hyperparameter configurations and train on half of the dataset for 50 epochs. We use $B = 4$ agents for RS, PBT and PB2, with $t_{\text{ready}}$ as 5 epochs. For ASHA we have the same maximum budget across all agents but begin with a population size of 16.

Table 2: Median best performing agent across 5 seeds. The best performing methods are bolded.

|  | RS | ASHA | PBT | PB2 |
|---|---|---|---|---|
| Test Accuracy | 84.43 | 88.85 | 87.20 | **89.10** |

In Table 7 we present the median best performing agent from five runs of each algorithm. We see that PB2 outperforms all other methods, including ASHA, which was specifically designed for SL problems [41]. This result indicates PB2 may be useful for a vast array of applications.

## 8   Experiment Details

For all experiments we set $\beta_t = c_1 + \log(c_2 t)$ with $c_1 = 0.2$ and $c_2 = 0.4$, as in the traffic speed data experiment from [9].

In Table 3, 5 & 7 and we show hyperparameters for the IMPALA, PPO and CIFAR experiments. In Table 8, 6 and 6 we show the bounds for the hyperparameters learned by PBT and PB2. All methods were initialized by randomly sampling from these bounds.

Table 3: IMPALA: Fixed

| Parameter | Value |
|---|---|
| Num Workers | 5 |
| Num GPUs | 0 |

Table 4: IMPALA: Learned

| Parameter | Value |
|---|---|
| Epsilon | $\{0.01, 0.5\}$ |
| Learning Rate | $\{10^{-3}, 10^{-5}\}$ |
| Entropy Coeff | $\{0.001, 0.1\}$ |

Table 5: PPO: Fixed

| Parameter | Value |
|---|---|
| Filter | $MeanStdFilter$ |
| SGD Iterations | 10 |
| Architecture | 32-32 |
| ready | $5 \times 10^4$ |

Table 6: PPO: Learned

| Parameter | Value |
|---|---|
| Batch Size | $\{1000, 60000\}$ |
| GAE $\lambda$ | $\{0.9, 0.99\}$ |
| PPO Clip $\epsilon$ | $\{0.1, 0.5\}$ |
| Learning Rate $\eta$ | $\{10^{-3}, 10^{-5}\}$ |

The model used for the CIFAR dataset was from: https://zhenye-na.github.io/2018/09/28/pytorch-cnn-cifar10.html. All experiments were run using a 32 core machine.

Table 7: CIFAR: Fixed

| Parameter | Value |
|---|---|
| Optimizer | Adam |
| Iterations | 50 |
| Architecture | 3 Conv Layers |
| ready | 5 |

Table 8: CIFAR: Learned

| Parameter | Value |
|---|---|
| Train Batch Size | $\{4, 128\}$ |
| Dropout-1 | $\{0.1, 0.5\}$ |
| Dropout-2 | $\{0.1, 0.5\}$ |
| Learning Rate | $\{10^{-3}, 10^{-4}\}$ |
| Weight Decay | $\{10^{-3}, 10^{-5}\}$ |
| Momentum | $\{0.8, 0.99\}$ |

# 9 Theoretical Results

We show the derivation for Lemma 1.

*Proof.* We have a reward at the starting iteration $F_1(x_1)$ as a constant that allows us to write the objective function as:

$$F_T(x_T) - F_1(x_1) = F_T(x_T) - F_{T-1}(x_{T-1}) + \cdots + F_3(x_3) - F_2(x_2) + F_2(x_2) - F_1(x_1) \quad (6)$$

Therefore, maximizing the left of Eq. (6) is equivalent to minimizing the cummulative regret as follows:

$$\max [F_T(x_T) - F_1(x_1)] = \max \sum_{t=1}^{T} F_t(x_t) - F_{t-1}(x_{t-1}) = \max \sum_{t=1}^{T} f_t(x_t) = \min \sum_{t=1}^{T} r_t(x_t)$$

where we define $f_t(x_t) = F_t(x_t) - F_{t-1}(x_{t-1})$, the regret $r_t = f_t(x_t^*) - f_t(x_t)$ and $f_t(x_t^*) := \max_{\forall x} f_t(x)$ is an unknown constant. $\square$

## 9.1 Convergence Analysis

We minimize the cumulative regret $R_T$ by sequentially suggesting an $\mathbf{x}_t$ to be used in each iteration $t$. We shall derive the upper bound in the cumulative regret and show that it asymptotically goes to zero as $T$ increases, i.e., $\lim_{T\to\infty} \frac{R_T}{T} = 0$. We make the following smoothness assumption to derive the regret bound of the proposed algorithm.

**Assumptions.** We will assume that the kernel $k$ is hold for some $(a, b)$ and all $L \geq 0$. The joint kernel satisfies for $k = 1, ..., K$,

$$\forall L \geq 0, t \leq \mathcal{T}, p(\sup \left| \frac{\partial f_t(\boldsymbol{\beta})}{\partial \boldsymbol{\beta}^{(k)}} \right| \geq L_t) \leq a e^{-(L_t/b)^2}. \quad (7)$$

These assumptions are achieved by using a time-varying kernel $k_{time}(t, t') = (1 - \omega)^{\frac{|t-t'|}{2}}$ [9] with the smooth functions [9]. For completeness, we restate Lemma 3, 4, 5, 6 from [61, 9], then present our new theoretical results in Lemma 7, 8, 9 and Theorem 10.

**Lemma 3** ([61]). *Let* $L_t = b\sqrt{\log 3da\frac{\pi_t}{\delta}}$ *where* $\sum_{t=1}^{T} \frac{1}{\pi_t} = 1$, *we have with probability* $1 - \frac{\delta}{3}$,

$$|f_t(\mathbf{x}) - f_t(\mathbf{x}')| \leq L_t ||\mathbf{x} - \mathbf{x}'||_1, \forall t, \mathbf{x}, \mathbf{x}' \in D. \quad (8)$$

**Lemma 4** ([61]). *We define a discretization* $D_t \subset D \subseteq [0, r]^d$ *of size* $(\tau_t)^d$ *satisfying* $||\mathbf{x} - [\mathbf{x}]_t||_1 \leq \frac{d}{\tau_t}, \forall \mathbf{x} \in D$ *where* $[\mathbf{x}]_t$ *denotes the closest point in* $D_t$ *to* $\mathbf{x}$. *By choosing* $\tau_t = \frac{t^2}{L_t d} = rdbt^2\sqrt{\log(3da\pi_t/\delta)}$, *we have*

$$|f_t(\mathbf{x}) - f_t([\mathbf{x}]_t)| \leq \frac{1}{t^2}.$$

**Lemma 5** ([61]). *Let* $\beta_t \geq 2\log\frac{3\pi_t}{\delta} + 2d\log\left(rdbt^2\sqrt{\log\frac{3da\pi_t}{2\delta}}\right)$ *where* $\sum_{t=1}^{T} \pi_t^{-1} = 1$, *then with probability at least* $1 - \frac{\delta}{3}$, *we have*

$$|f_t(\mathbf{x}_t) - \mu_t(\mathbf{x}_t)| \leq \sqrt{\beta_t}\sigma_t(\mathbf{x}_t), \forall t, \forall \mathbf{x} \in \mathcal{D}.$$

*Proof.* We note that conditioned on the outputs $(y_1, ..., y_{t-1})$, the sampled points $(\mathbf{x}_1, ..., \mathbf{x}_t)$ are deterministic, and $f_t(\mathbf{x}_t) \sim \mathcal{N}\left(\mu_t(\mathbf{x}_t), \sigma_t^2(\mathbf{x}_t)\right)$. Using Gaussian tail bounds [64], a random variable $f \sim \mathcal{N}(\mu, \sigma^2)$ is within $\sqrt{\beta}\sigma$ of $\mu$ with probability at least $1 - \exp\left(-\frac{\beta}{2}\right)$. Therefore, we first claim that if $\beta_t \geq 2\log\frac{3\pi_t}{\delta}$ then the selected points $\{\mathbf{x}_t\}_{t=1}^{T}$ satisfy the confidence bounds with probability at least $1 - \frac{\delta}{3}$

$$|f_t(\mathbf{x}_t) - \mu_t(\mathbf{x}_t)| \leq \sqrt{\beta_t}\sigma_t(\mathbf{x}_t), \forall t. \quad (9)$$

This is true because the confidence bound for individual $\mathbf{x}_t$ will hold with probability at least $1 - \frac{\delta}{3\pi_t}$ and taking union bound over $\forall t$ will lead to $1 - \frac{\delta}{3}$.

We show above that the bound is applicable for the selected points $\{\mathbf{x}_t\}_{t=1}^T$. To ensure that the bound is applicable for all points in the domain $D_t$ and $\forall t$, we can set $\beta_t \geq 2\log\frac{3|D_t|\pi_t}{\delta}$ where $\sum_{t=1}^T \pi_t^{-1} = 1$ e.g., $\pi_t = \frac{\pi^2 t^2}{6}$

$$p(|f_t(\mathbf{x}_t) - \mu_t(\mathbf{x}_t)| \leq \sqrt{\beta_t}\sigma_t(\mathbf{x}_t) \geq 1 - |D_t| \sum_{t=1}^T \exp(-\beta_t/2) = 1 - \frac{\delta}{3.} \tag{10}$$

By discretizing the domain $D_t$ in Lem. 4, we have a connection to the cardinality of the domain that $|D_t| = (\tau_t)^d = \left(rdbt^2\sqrt{\log(3da\pi_t/\delta)}\right)^d$. Therefore, we need to set $\beta_t$ such that both conditions in Eq. (9) and Eq. (10) are satisfied. We simply take a sum of them and get $\beta_t \geq 2\log\frac{3\pi_t}{\delta} + 2d\log\left(rdbt^2\sqrt{\log\frac{3da\pi_t}{2\delta}}\right)$. $\square$

We use $TB$ to denote the batch setting where we will run the algorithm over $T$ iterations with a batch size $B$. The mutual information is defined as $\tilde{\mathbf{I}}(f_{TB}; y_{TB}) = \frac{1}{2}\log\det\left(\mathbf{I}_{TB} + \sigma_f^{-2}\tilde{K}_{TB}\right)$ and the maximum information gain is as $\tilde{\gamma}_T := \max\tilde{\mathbf{I}}(\mathbf{f}_{TB}; \mathbf{y}_{TB})$ where $\mathbf{f}_{TB} := f_{TB}(\mathbf{x}_{TB}) = (f_{t,b}(\mathbf{x}_{t,b}), ..., f_{T,B}(\mathbf{x}_{T,B})), \forall b = 1....B, \forall t = t...T$ for the time variant GP $f$. Using the result presented in [9], we can adapt the bound on the time-varying information gain into the parallel setting using a population size of $B$ below.

**Lemma 6.** *(adapted from [9] with a batch size $B$) Let $\omega$ be the forgetting-remembering trade-off parameter and consider the kernel for time $1 - K_{time}(t, t') \leq \omega|t - t'|$, we bound the maximum information gain that*

$$\tilde{\gamma}_{TB} \leq \left(\frac{T}{\tilde{N} \times B} + 1\right)\left(\gamma_{\tilde{N} \times B} + \sigma_f^{-2}\left[\tilde{N} \times B\right]^3 \omega\right).$$

**Uncertainty Sampling (US).** We next derive an upper bound over the maximum information gain obtained from a batch $\mathbf{x}_{t,b}, \forall b = 1, ..., B$. In other words, we want to show that the information gain by our chosen points $\mathbf{x}_{t,b}$ will not go beyond the ones by maximizing the uncertainty. For this, we define an uncertainty sampling (US) scheme which fills in a batch $\mathbf{x}_{t,b}^{US}$ by maximizing the GP predictive variance. Particularly, at iteration $t$, we select $\mathbf{x}_{t,b}^{US} = \arg\max_{\mathbf{x}} \sigma_t(\mathbf{x} \mid D_{t,b-1}), \forall b \leq B$ and the data set is augmented over time to include the information of the new point, $D_{t,b} = D_{t,b-1} \cup \mathbf{x}_{t,b}^{US}$. We note that we use $\mathbf{x}_{t,b}^{US}$ to derive the upper bound, but this is not used by our PB2 algorithm.

**Lemma 7.** *Let $\mathbf{x}_{t,b}^{PB2}$ be the point chosen by our algorithm and $\mathbf{x}_{t,b}^{US}$ be the point chosen by uncertainty sampling (US) by maximizing the GP predictive variance $\mathbf{x}_{t,b}^{US} = \arg\max_{\mathbf{x} \in D} \sigma_t(\mathbf{x} \mid D_{t,b-1}), \forall b = 1, ...B$ and $D_{t,b} = D_{t,b-1} \cup \mathbf{x}_{t,b}$. We have*

$$\sigma_{t+1,1}\left(\mathbf{x}_{t+1,1}^{PB2}\right) \leq \sigma_{t+1,1}\left(\mathbf{x}_{t+1,1}^{US}\right) \leq \sigma_{t,b}\left(\mathbf{x}_{t,b}^{US}\right), \forall t \in \{1, ..., T\}, \forall b \in \{1, ...B\}.$$

*Proof.* The first inequality is straightforward that the point chosen by uncertainty sampling will have the highest uncertainty $\sigma_{t+1,1}\left(\mathbf{x}_{t+1,1}^{PB2}\right) \leq \sigma_{t+1,1}\left(\mathbf{x}_{t+1,1}^{US}\right) = \arg\max_{\mathbf{x}} \sigma_t(\mathbf{x} \mid D_{t,b-1})$.

The second inequality is obtained by using the principle of "information never hurts" [36], we know that the GP uncertainty for all locations $\forall\mathbf{x}$ decreases with observing a new point. Therefore, the uncertainty at the future iteration $\sigma_{t+1}$ will be smaller than that of the current iteration $\sigma_t$, i.e., $\sigma_{t+1,b}\left(\mathbf{x}_{t+1,b}^{US}\right) \leq \sigma_{t,b}\left(\mathbf{x}_{t,b}^{US}\right), \forall b \leq B, \forall t \leq T$. We thus conclude the proof $\sigma_{t+1,1}\left(\mathbf{x}_{t+1,1}^{US}\right) \leq \sigma_{t,b}\left(\mathbf{x}_{t,b}^{US}\right), \forall t \in \{1, ..., T\}, \forall b \in \{1, ...B\}$. $\square$

**Lemma 8.** *The sum of variances of the points selected by the our PB2 algorithm $\sigma()$ is bounded by the sum of variances by uncertainty sampling $\sigma^{US}()$. Formally, w.h.p., $\sum_{t=2}^T \sigma_{t,1}(\mathbf{x}_{t,1}) \leq \frac{1}{B}\sum_{t=1}^T\sum_{b=1}^B \sigma_{t,b}\left(\mathbf{x}_{t,b}^{US}\right)$.*

*Proof.* By the definition of uncertainty sampling in Lem. 7, we have $\sigma_{t+1,1}(\mathbf{x}_{t+1,1}) \leq \sigma_{t,b}\left(\mathbf{x}_{t,b}^{\text{US}}\right), \forall t \in \{1,...,T\}, \forall b \in \{2,...B\}$ and $\sigma_{t,1}(\mathbf{x}_{t,1}) \leq \sigma_{t,1}(\mathbf{x}_{t,1}^{\text{US}})$ where $\mathbf{x}_{t,1}$ is the point chosen by our PB2 and $\mathbf{x}_{t,1}^{\text{US}}$ is from uncertainty sampling. Summing all over $B$, we obtain

$$\sigma_{t,1}(\mathbf{x}_{t,1}) + (B-1)\sigma_{t+1,1}(\mathbf{x}_{t+1,1}) \leq \sigma_{t,1}(\mathbf{x}_{t,1}^{\text{US}}) + \sum_{b=2}^{B} \sigma_{t,b}(\mathbf{x}_{t,b}^{\text{US}})$$

$$\sum_{t=1}^{T} \sigma_{t,1}(\mathbf{x}_{t,1}) + (B-1)\sum_{t=1}^{T}\sigma_{t+1,1}(\mathbf{x}_{t+1,1}) \leq \sum_{t=1}^{T}\sum_{b=1}^{B} \sigma_{t,b}(\mathbf{x}_{t,b}^{\text{US}}) \qquad \text{by summing over } T$$

$$\sum_{t=2}^{T} \sigma_{t,1}(\mathbf{x}_{t,1}) \leq \frac{1}{B}\sum_{t=1}^{T}\sum_{b=1}^{B} \sigma_{t,b}(\mathbf{x}_{t,b}^{\text{US}}).$$

The last equation is obtained because of $\sigma_{1,1}(\mathbf{x}_{1,1}) \geq 0$ and $(B-1)\sigma_{T+1,1}(\mathbf{x}_{T+1,1}) \geq 0$. $\qquad \square$

**Lemma 9.** *Let $C_1 = \frac{32}{\log\left(1+\sigma_f^{-2}\right)}$, $\sigma_f^2$ be the measurement noise variance and $\tilde{\gamma}_{TB} := \max \tilde{\mathbf{I}}$ be the maximum information gain of time-varying kernel, we have $\sum_{t=1}^{T}\sum_{b=1}^{B}\sigma_{t,b}^2(\mathbf{x}_{t,b}^{\text{US}}) \leq \frac{C_1}{16}\tilde{\gamma}_{TB}$ where $\mathbf{x}_{t,b}^{US}$ is the point selected by uncertainty sampling (US).*

*Proof.* We show that $\sigma_{t,b}^2(\mathbf{x}_{t,b}^{\text{US}}) = \sigma_f^2\left(\sigma_f^{-2}\sigma_{t,b}^2(\mathbf{x}_{t,b}^{\text{US}})\right) \leq \sigma_f^2 C_2 \log\left(1+\sigma_f^{-2}\sigma_{t,b}^2\left(\mathbf{x}_{t,b}^{\text{US}}\right)\right), \forall b \leq B, \forall t \leq T$ where $C_2 = \frac{\sigma_f^{-2}}{\log\left(1+\sigma_f^{-2}\right)} \geq 1$ and $\sigma_f^2$ is the measurement noise variance. We have the above inequality because $s^2 \leq C_2 \log\left(1+s^2\right)$ for $s \in \left[0, \sigma_f^{-2}\right]$ and $\sigma_f^{-2}\sigma_{t,b}^2\left(\mathbf{x}_{t,b}^{\text{US}}\right) \leq \sigma^{-2}k\left(\mathbf{x}_{t,b}^{\text{US}}, \mathbf{x}_{t,b}^{\text{US}}\right) \leq \sigma_f^{-2}$. We then use Lemma 5.3 of [14] to have the information gain over the points chosen by a time-varying kernel $\tilde{\mathbf{I}} = \frac{1}{2}\sum_{t=1}^{T}\sum_{b=1}^{B}\log\left(1+\sigma_f^{-2}\sigma_{t,b}^2\left(\mathbf{x}_{t,b}^{\text{US}}\right)\right)$. Finally, we obtain

$$\sum_{t=1}^{T}\sum_{b=1}^{B}\sigma_{t,b}^2(\mathbf{x}_{t,b}^{\text{US}}) \leq \sigma_f^2 C_2 \sum_{t=1}^{T}\sum_{b=1}^{B}\log\left(1+\sigma_f^{-2}\sigma_{t,b}^2\left(\mathbf{x}_{t,b}^{\text{US}}\right)\right) = 2\sigma_f^2 C_2 \tilde{\mathbf{I}} = \frac{C_1}{16}\tilde{\gamma}_{TB}$$

where $C_1 = \frac{2}{\log\left(1+\sigma_f^{-2}\right)}$ and $\tilde{\gamma}_{TB} := \max \tilde{\mathbf{I}}$ is the definition of maximum information gain given by $T \times B$ data points from a GP for a specific time-varying kernel. $\qquad \square$

**Theorem 10.** *Let the domain $\mathcal{D} \subset [0,r]^d$ be compact and convex where $d$ is the dimension and suppose that the kernel is such that $f \sim GP(0,k)$ is almost surely continuously differentiable and satisfies Lipschitz assumptions for some $a, b$. Fix $\delta \in (0,1)$ and set $\beta_T = 2\log\frac{\pi^2 T^2}{2\delta} + 2d\log rdbT^2\sqrt{\log\frac{da\pi^2 T^2}{2\delta}}$. Defining $C_1 = 32/\log(1+\sigma_f^2)$, the PB2 algorithm satisfies the following regret bound after $T$ time steps:*

$$R_{TB} = \sum_{t=1}^{T} f_t(\mathbf{x}_t^*) - f_t(\mathbf{x}_t) \leq \sqrt{C_1 T \beta_T \left(\frac{T}{\tilde{N}B}+1\right)\left(\gamma_{\tilde{N}B} + \left[\tilde{N}B\right]^3 \omega\right)} + 2$$

*with probability at least $1-\delta$, the bound is holds for any $\tilde{N} \in \{1,...,T\}$ and $B \ll T$.*

*Proof.* Let $\mathbf{x}_t^* = \arg\max_{\forall \mathbf{x}} f_t(\mathbf{x})$ and $\mathbf{x}_{t,b}$ be the point chosen by our algorithm at iteration $t$ and batch element $b$, we define the (time-varying) instantaneous regret as $r_{t,b} = f_t(\mathbf{x}_t^*) - f_t(\mathbf{x}_{t,b})$ and the (time-varying) batch instantaneous regret over $B$ points is as follows

$$r_t^B = \min_{b \leq B} r_{t,b} = \min_{b \leq B} f_t(\mathbf{x}_t^*) - f_t(\mathbf{x}_{t,b}), \forall b \leq B$$

$$\leq f_t(\mathbf{x}_t^*) - f_t(\mathbf{x}_{t,1}) \leq \mu_t(\mathbf{x}_t^*) + \sqrt{\kappa_t}\sigma_t(\mathbf{x}_t^*) + \frac{1}{t^2} - f_t(\mathbf{x}_{t,1}) \qquad \text{by Lem. 4}$$

$$\leq \mu_t(\mathbf{x}_{t,1}) + \sqrt{\kappa_t}\sigma_t(\mathbf{x}_{t,1}) + \frac{1}{t^2} - f_t(\mathbf{x}_{t,1}) \leq 2\sqrt{\kappa_t}\sigma_t(\mathbf{x}_{t,1}) + \frac{1}{t^2} \qquad (11)$$

where we have used the property that $\mu_t(\mathbf{x}_{t,1}) + \sqrt{\beta_t}\sigma_t(\mathbf{x}_{t,1}) \geq \mu_t(\mathbf{x}_t^*) + \sqrt{\beta_t}\sigma_t(\mathbf{x}_t^*)$ by the definition of selecting $\mathbf{x}_{t,1} = \arg\max_{\mathbf{x}} \mu_t(\mathbf{x}) + \sqrt{\beta_t}\sigma_t(\mathbf{x})$. Next, we bound the cumulative batch regret as

$$R_{TB} = \sum_{t=1}^{T} r_t^B \leq \sum_{t=1}^{T} \left( 2\sqrt{\kappa_t}\sigma_t(\mathbf{x}_{t,1}) + \frac{1}{t^2} \right) \qquad \text{by Eq. (11)}$$

$$\leq 2\sqrt{\kappa_T}\sigma_1(\mathbf{x}_{1,1}) + \frac{2\sqrt{\kappa_T}}{B}\sum_{t=1}^{T}\sum_{b=1}^{B}\sigma_{t,b}\left(\mathbf{x}_{t,b}^{\mathrm{US}}\right) + \sum_{t=1}^{T}\frac{1}{t^2} \qquad \text{by Lem. 7 and } \kappa_T \geq \kappa_t, \forall t \leq T$$

$$\leq \frac{4\sqrt{\kappa_T}}{B}\sum_{t=1}^{T}\sum_{b=1}^{B}\sigma_{t,b}\left(\mathbf{x}_{t,b}^{\mathrm{US}}\right) + \sum_{t=1}^{T}\frac{1}{t^2} \qquad (12)$$

$$\leq \frac{4}{B}\sqrt{\kappa_T \times TB \sum_{t=1}^{T}\sum_{b=1}^{B}\sigma_{t,b}^2(\mathbf{x}_{t,b}^{\mathrm{US}})} + 2 \leq \sqrt{C_1 \frac{T}{B}\kappa_T \tilde{\gamma}_{TB}} + 2 \qquad (13)$$

$$\leq \sqrt{C_1 \frac{T}{B}\kappa_T \left( \frac{T}{\tilde{N}B} + 1 \right)\left( \gamma_{\tilde{N}B} + \frac{1}{\sigma_f^2}\left[ \tilde{N}B \right]^3 \omega \right)} + 2 \qquad (14)$$

where $C_1 = 32/\log(1+\sigma_f^2)$, $\mathbf{x}_{t,b}^{\mathrm{US}}$ is the point chosen by uncertainty sampling – used to provide the upper bound in the uncertainty. In Eq. (12), we take the upper bound by considering two possible cases: either $\sigma_1(\mathbf{x}_{1,1}) \geq \frac{1}{B}\sum_{t=1}^{T}\sum_{b=1}^{B}\sigma_{t,b}\left(\mathbf{x}_{t,b}^{\mathrm{US}}\right)$ or $\frac{1}{B}\sum_{t=1}^{T}\sum_{b=1}^{B}\sigma_{t,b}\left(\mathbf{x}_{t,b}^{\mathrm{US}}\right) \geq \sigma_1(\mathbf{x}_{1,1})$. It results in $\frac{2}{B}\sum_{t=1}^{T}\sum_{b=1}^{B}\sigma_{t,b}\left(\mathbf{x}_{t,b}^{\mathrm{US}}\right) \geq \frac{1}{B}\sum_{t=1}^{T}\sum_{b=1}^{B}\sigma_{t,b}\left(\mathbf{x}_{t,b}^{\mathrm{US}}\right) + \sigma_1(\mathbf{x}_{1,1})$. In Eq. (13) we have used $\sum_{t=1}^{\infty}\frac{1}{t^2} \leq \pi^2/6 \leq 2$ and $||z||_1 \leq \sqrt{T}||z||_2$ for any vector $z \in \mathcal{R}^T$. In Eq. (14), we utilize Lem. 6.

Finally, given the squared exponential (SE) kernel defined, $\gamma_{\tilde{N}B}^{SE} = \mathcal{O}\left( \left[ \log \tilde{N}B \right]^{d+1} \right)$, the bound is $R_{TB} \leq \sqrt{C_1 \frac{T}{B}\beta_T \left( \frac{T}{\tilde{N}B} + 1 \right)\left( (d+1)\log\left( \tilde{N}B \right) + \frac{1}{\sigma_f^2}\left[ \tilde{N}B \right]^3 \omega \right)} + 2$ where $\tilde{N} \leq T$ and $B \ll T$. $\qquad \square$

In our time-varying setting, if the time-varying function is highly correlated, i.e., the information between $f_1(.)$ and $f_T(.)$ does not change significantly, we have $\omega \to 0$ and $\tilde{N} \to T$. Then, the regret bound grows sublinearly with the number of iterations $T$, i.e., $\lim_{T \to \infty} \frac{R_{TB}}{TB} = 0$. This bound suggests that the gap between $f_t(\mathbf{x}_t)$ and the optimal $f_t(\mathbf{x}_t^*)$ vanishes asymptotically using PB2. In addition, our regret bound is tighter and better with increasing batch size $B$.

On the other hand in the worst case, if the time-varying function is not correlated, such as $\tilde{N} \to 1$ and $\omega \to 1$, then PB2 achieves the linear regret [9].