[Reviews · NeurIPS 2020]

Review 1

Summary and Contributions: ============= Post Rebuttal ============= I thank the authors for answering my questions and the extra work they put into the rebuttal. My major concern was the empirical evaluation of the paper and the authors improved that during the rebuttal. While I am still not entirely convinced about the practical gains of the method, I do think the paper represents a step towards model-based online hyperparameter optimization and lays the foundation for interesting future work. I am willing to increase my score (5->6) and recommend acceptance. For the camera ready copy, I suggest to add the following plots (could be also in the supplement material): - add a plot that shows the performance over wall-clock time - add Figure 1 from the rebuttal but over time steps to show the anytime performance of PB2 vs BO ======================================= The paper presents a new hyperparameter optimization method to adjust the hyperparameters of reinforcement learning algorithms online. Inspired by population based training, which maintains a population of neural networks and randomly permutes the hyperparameters of the poorly performing networks in the population, this paper proposes to use Bayesian optimization to sample new candidates for the population. The resulting algorithm PB2 achieves a better sample efficiency on reinforcement learning benchmarks compared to the original population based training as well as other random search based methods, such as ASHA, that treat hyperparameter optimization more from a black box perspective.

Strengths: The main strength of the paper is to present, as far as I am aware of, the first online Bayesian hyperparameter optimization method, which tackles a problem of high practical relevance. As the paper highlights, this might be of particularly importance for reinforcement learning algorithms which are known to be notoriously brittle against their hyperparameters.

Weaknesses: I think the main weakness of the paper is the empirical evaluation of the method (see Section Correctness for more details), which raises the question whether they are any benefits of the proposed method compared to other much simpler existing methods in practice.

Correctness: Looking at Figure 2 it seems that, given the large uncertainty bands, PB2 often performs only slightly better than the much simpler ASHA method. I am also missing a comparison to standard Bayesian optimization, which obviously is not able to adapt hyperparameters on the fly, but, given the strong performance of ASHA, this might be actually not as important as the paper suggests. I would have also liked to see the same plots as in Figure 2 just with wall-clock time on the x-axis. I am concerned that, while the proposed method performs better in terms of sample efficiency, it might underperform in terms of wall-clock time due to the computational overhead of the GP method, which is arguably more relevant in practice.

Clarity: The paper is well written and easy to follow. There are just a few points that should be clarified: - Equation 4: What exactly is f that is multiplied to g_{t+1}? Could that be a typo? - Section 3.1: it says that if \omega = 0 it returns to GP-UCB. Why GP-UCB? - What is the reduction factor \eta of ASHA?

Relation to Prior Work: The method seem quite related to BOHB (Falkner et al.) which combines Hyperband with Bayesian optimization, and the most notably difference seems to be that hyperparameter are changed on the fly instead of evaluating on smaller budgets first. However, given the already strong performance of ASHA, I think the empirical evaluation would be more convincing if it contains BOHB as additional baseline.

Reproducibility: Yes

Additional Feedback: I suggest the authors to improve the paper by addressing the following points: - add a comparison to at least standard Bayesian optimization or, even better, to more sophisticated methods such as for example BOHB - plot performance not only over time steps but also over wall-clock time Typos: - Abstract second sentence: hyperaameters -> hyperparameters


Review 2

Summary and Contributions: The paper provides a more efficient alternative for candidate selection in PBT using Gaussian processes, making it a more viable hyperparameter optimization method for applications without high compute resources. The model based candidate selection not only accelerates the process and requires a smaller population than the original PBT, the authors are also able to bound the regret for a given time step which was not possible with the original exploration strategy. Furthermore, the use of bandits to select possible hyperparameter values mitigates a key weakness of the original PBT algorithm, namely a significantly worse performance when hyperparameter bounds are not specified correctly. These contributions significantly improve upon PBT, as shown thoroughly in the experiment section, while also providing a theoretical foundation for its efficiency.

Strengths: PB2 is an improvement over PBT in many ways, most significantly in that it greatly improves the usability of PBT with more modest computing resources. These improvements are demonstrated in well-designed experiments that cover possible applications of PB2 well. Experiments include on- and off-policy reinforcement learning using state-of-the-art learning algorithms and also supervised learning. In the main application area of reinforcement learning, experiments span a variety of environments and the authors provide good comparisons in the on-policy case. The regret bound given in this paper provides a guarantee for the results shown in the experimental section, which is not often available for hyperparameter optimization methods.

Weaknesses: In the experiments, the most important comparison is of course to PBT. It makes sense for this paper to compare PBT and PB2 using the same budget, however it would also have been interesting to see a comparison to full-scale PBT. I believe the current results shown are a compelling argument to use PB2 over PBT with the possibility to run only a few agents in parallel. I would also have liked to see how much PBT is improved with more resources and how PB2 compares in that case. The experiments presented here do not show the full picture in that regard. Random search and ASHA are fine baselines, but there are other comparisons that might have been more interesting. A population based BO methods, for example, are mentioned in the related work and are similar enough to warrant the comparison. BO methods themselves would also represent state-of-the-art hyperparameter optimization better than random search. Lastly, a variation of PB2 with random search instead of gaussian processes would have been a nice baseline to really see the improvement from the performance model. In fact the changed settings, shown in Figure 4, seem to be fairly random and do not follow an obvious pattern, which gives even more reason to compare against random search. The supervised learning section of the experiments is a good idea in principle, the execution is lacking, however. The used architecture as well as the results are not close to state-of-the-art performance at all. This is not a good example of successful hyperparameter optimization for supervised learning and weakens the experimental section as a whole. I’m also missing a discussion of the assumptions of Theorem 2. How likely is it that the space is convex and und which cases B << T hold?

Correctness: Overall, the approach and the benchmarking methods are fine. However, Equation 4 is not totally clear to me. What is fg_{t+1}(x). Is f somehow applied to g?

Clarity: The paper is clear overall and makes an effort to be accessible for the most part. It felt that this effort for clarity, however, did not always extend to the theoretical results which are not unclear but more compressed compared to the rest of the paper.

Relation to Prior Work: The related work section takes different related fields into account and shows clearly where PB2 stands in comparison.

Reproducibility: Yes

Additional Feedback: The algorithm does not explicitly mark its hyperparameters (e.g. t_ready). Explicitly including this in the algorithm’s pseudocode would improve readability. In the supplementary material in line 503, you say “kernel k is hold for some (a, b)”. It reads like a mistake to me, if not it might simply not be expressed very clearly. In Figure 1, describing the performance bar simply with “high performance” and the symbol in the legend, it is not immediately clear that more orange in the bar means better performance. This could maybe be expressed better. >>>>>>>>>>>>>>>>>>>>>>>>>>> Comments after Rebuttal >>>>>>>>>>>>>>>>>>>>>>>>>>> Thank your for the rebuttal. It helped to better understand the paper and the results. Some few comments (which are also based on the other reviews and the internal discussion): (I) Actually, I would love to see benchmarks with larger networks, which are sota. But I can fully understand that not all labs can afford such benchmarks. Overall, I think the benchmark on the small CNN is still a weak point of the paper. I would propose to really focus on the RL benchmarks and either drop the CNN benchmarks or move it into the appendix. (II) The rebuttal gives a first glimpse that your approach works well under the setting you chose. Nevertheless, I believe it would be easy to construct a benchmark where BO would outperform PB2. This is not a bad thing, but the paper should be more explicit under which assumptions PB2 will outperform approaches such as BO and discuss the corresponding limitations. (III) I would still love to see PB2 with even larger population sizes, since I could guess that it will not scale nicely --- I hope that it will at least perform as well as parallelized random search at some point. I fully understand that your goal was to address small population sizes (which is very relevant in practice), but in view of studying the limitations of the approach, I would be interested to see it break.


Review 3

Summary and Contributions: This work proposes a novel hyperparameter optimization algorithm, called PB2, that improves on the older population based training (PBT) algorithm, by framing the optimization problem as a gaussian process optimization in the bandit setting. PB2 is empirically shown to outperform PBT and other competitive baselines on reinforcement learning and image classification tasks. The paper also proves that with high probability PB2 can achieve sublinear regret.

Strengths: - The paper is clear and well written. - The experiments in section 5 clearly show that PB2 outperforms or is at least competitive with other baselines.

Weaknesses: - The proposed algorithm does not present significant novelty over prior works. - The experimental section can be strengthened. I write more on those weaknesses in the feedback section.

Correctness: The claims and empirical methodology are sound.

Clarity: The paper is well written.

Relation to Prior Work: The paper clearly discusses prior works and highlights this work's differentiating features.

Reproducibility: Yes

Additional Feedback: The empirical results shown in the RL setting are very good, especially in the "low budget" regime with only 4 agents. However, as I mention above I have a few concerns about this work. The true novelty in this work is in recognizing that the GP bandits from [9] can be used to improve the "explore" function of the PBT algorithm in [33] (which appears on line 9 of algorithm 1 in [33]). The theoretical result presented in Theorem 2 is a direct extension of Theorem 4.3 of [9] to the case of multiple agents (i.e. B>1). I have not read the proof of theorem 2 in detail, however, it would be helpful if the authors can point out distinguishing features in their proof from the proof presented in appendix C of [9]. In section 5.3 what hyperparameters are being optimized? I believe the paper can be strengthened by showing more results in the supervised learning setting, perhaps by comparing PB2 to PBT and ASHA on larger datasets using larger networks, perhaps even by looking for optimal network architectures. If time allows, please comment on this in your rebuttal. Those are my two biggest concerns, I would be happy to raise my score if the authors can address them. I also found two minor typos: -line 153: *equivalence -line 144: It would help to have parentheses around the logarithms. %%%%%%POST REBUTTAL%%%%%%%%%%%% I'd like to thank the authors for their strong rebuttal. After discussing internally with other reviewers, I am increasing my score to a 6. For the camera-ready version, I would still like to ask the reviewers to either strengthen section 5.3 (experiments on supervised learning tasks) with more experiments, or move the entire section tot he appendix. [9] I. Bogunovic, J. Scarlett, and V. Cevher. Time-Varying Gaussian Process Bandit Optimization. In Proceedings of the 19th International Conference on Artificial Intelligence and Statistics, 2016. [33] M. Jaderberg, V. Dalibard, S. Osindero, W. M. Czarnecki, J. Donahue, A. Razavi, O. Vinyals, T. Green, I. Dunning, K. Simonyan, C. Fernando, and K. Kavukcuoglu. Population based training of neural networks. CoRR, abs/1711.09846, 2017.


Review 4

Summary and Contributions: This paper proposes a new method to automatically tune hyperparameters for the RL domain, where the method is based on a hybridization of population based training (PBT) and Bayesian optimization (BO). Theoretical analyses was provided to prove that the defined regret is bounded sublinearly using the proposed method. Experiments for on-policy RL, off-policy RL, and SL show that the proposed PB2 algorithm outperforms classic PBT and a baseline algorithm (ASHA). The main benefits that set PB2 apart from both PBT and BO are: 1) By using BO instead of hyperparameter perturbation when replacing the worst agents' hyperparameters, PB2 traverses the hyperparameter search space more efficiently than PBT (which uses random and greedy exploration). 2) The PBT portion of PB2 allows hyperparameter scheduling (dynamic changes to hyperparmeters as training progresses). 3) PB2 involves a single training run as opposed to finding an optimal set of hyperparameters over multiple runs.

Strengths: The inclusion of a theoretical basis for PB2 works in favor of the paper's soundness. The experiments cover both on and off policy settings in RL, with an extension into SL to examine generality of PB2.

Weaknesses: For the experiments, only B = 4 and 8 were tried, where for the on-policy setting the benefits of PB2 over PBT diminishes. While I understand one of the central goals of this paper is to propose a method that enables efficient hyperparameter tuning for research teams with limited resources, I would've still liked to see more experiments for larger B values, if only to facilitate a fuller picture of PB2's usefulness in relation to previous methods and problem settings given more resources. Likewise, B is also restricted to 4 for both the off-policy and SL experiments.

Correctness: The experiments are sound and I appreciate the variety of experiments in different problem domains. I have trouble following the theoretical portions, and will defer judgement on those sections and leave that for my fellow reviewers.

Clarity: Overall the paper is well-organized, with minor typos which I will list below.

Relation to Prior Work: The related work section is clear and the differences between PB2 and previous methods are listed.

Reproducibility: Yes

Additional Feedback: Minor mistakes: Abstract, line 4: hyperpaameters -> hyperparameters Figure 1 caption: it's weights are replaced -> its weights are replaced line 153: equivalent -> equivalence line 160: why -> while line 201: a similar to -> a similar method to ================== Post rebuttal =================== The authors have addressed my main concern and I have no further comments.

[Author Response · NeurIPS 2020]

We would like to thank the reviewers for their time. We want to reiterate our primary objective is to **make PBT more**
**efficient**, such that it can be **effective with a small computational budget**. This was motivated by our own frustration
with brittle RL hyperparameters, without the budget to run PBT. It is great to see all reviewers got this: R1 and R2
both saw the benefits in RL, saying this "might be of particularly importance for reinforcement learning algorithms",
and "empirical results shown in the RL setting are very good" respectively. Meanwhile R2 noted the improvement in
particular vs. PBT, and R4 appreciated both the theoretical results and variety of experiments. We now seek to clarify
all concerns, and hope this is sufficient to consider raising scores.

Figure 1: BipedalWalker, best rewards.

**BO Comparison R1 R2** We tested vanilla BO (sequential, EI, RatQuad) on the
BipedalWalker task. We used 500k samples for each trial, to have sufficient
number of evaluations in 4M and 8M timesteps (to compare vs. $B \in \{4, 8\}$). In
Fig 1 we see BO performs poorly with the 4M budget, and for the 8M budget still
underperforms PB2. This sequential method should be superior in performance
to batch BO methods, since each trial is completed with full knowledge of
all previous trials. The improvement therefore represents gains from updating
hyperparameters on the fly (also see Fig 5 d) of Jaderberg et al. 2017). We also
tried to compare against BOHB, however the ray tune version does not work for
RL. We do not think it is fair to compare codebases for RL (see: Engstrom, ICLR 2020), so will seek to resolve this in
time for the CRC. In the ASHA paper they outperform BOHB, so we feel this is a strong non-PBT baseline.

**Larger Population Sizes R2 R4** We agree it is important to assess PB2 with $B > 8$. To
answer this, we ran the BipedalWalker experiment with $B = 16$. As we see in Figure 2,
both PBT and PB2 achieve optimal rewards ($> 300$), but PB2 is still more efficient. We
hope this provides confidence in our method's effectiveness with more resources.

Figure 2: BipedalWalker, median, 5 seeds, $B = 16$.

**Larger Supervised Experiments R2 R3** We included the SL experiment to show our
method generalizes beyond RL, as noted by R4. The relative performance of the hyper-
parameter algorithms should be agnostic to the architecture, and PB2 outperforms other
competitive baselines **across five seeds**. We used a medium sized CNN, as this was the
most powerful model we could realistically train with our compute. Larger networks are prohibitively expensive for
our lab. Instead, we allocated our resources on larger RL settings such as Atari (which are more CPU intensive). For
reference, a similarly accurate CNN model was used for the first set of experiments in ASHA (as a POC). The main
result in BOHB was a CNN on CIFAR-10, but they used 19 workers, each with 2 GPUs. However, their RL experiments
were toy. In light of this, we think expecting SOTA SL experiments on a paper focused on RL is unreasonable.

**What is $fg_{t+1}(x)$? R1 R2** It should be written as $g_{t+1}(x)$, without $f$. Thank you for catching this.

Next we address individual comments in more detail.

**R1**: **Wall-clock comparison** For RL, the GP-bandit step takes a trivial amount of time compared to querying a simulator
or computing gradients, given we only have $< 10$ hyperparameters we have a tiny dataset. If we have vast computational
resources and we wish to optimize $\gg 10$ hyperparameters for $B > 20$, it may be required to use more scalable GP
methods. However, *this is not the problem we are trying to solve*. **PB2 only slightly outperforming ASHA** The main
purpose of this paper is to improve PBT, and our main goal is to show this. In addition, we also show PB2 outperforms
state of the art (ASHA). **Sec 3.1** To be precise, if (1) $\omega = 0$ and (2) the number of parallel agent $B = 1$, then our model
reduces to GP-UCB. The reduction factor is 3, this is the default in the ray tune implementation.

**R2**: Thank you very much for your comments, we are glad you feel this is an improvement over PBT, in particular
making it more usable, which was always our goal. We dont assume the convexity of the function. Instead, we make a
mild condition on the compact and convex input space so that the input space is continuous and not segmented, e.g.,
the case of non-convexity. This mild assumption is popular in GP bandit literature (see Srinivas et al, 2010). We
always have $B \ll T$ because the number of update step $T$ typically goes beyond thousands to millions while the
batch size $B = 4, 8, 16...$ is much smaller and limited to the parallel facility we have. **Hyperparameter Schedules** We
disagree the hyperparameters are random, for the learning rate in particular, you can see the GP balances exploration
and exploitation by either selecting points from a single mode or right at the boundaries.

**R3**: **Distinguishing features of our proof** Our proof makes two significant extensions beyond TV-GP-UCB (Bugonovic
et al, 2016). First, we improve the bound from C.2 from $\tilde{N}^3$ to $\tilde{N}^{2.5}$ (smaller is tighter and better). Second, we extend the
regret bound to the batch setting including the new Lemma 8,9,10 (in the Appendix). **Section 5.3** The hyperparameters
and ranges are shown in Table 9, in the Appendix (top of p14). **Larger datasets using larger networks** See above: we
hope this is not the primary reason for rejecting our work. **Optimal network architectures**, given recent works using
BO for NAS we think this is a logical next step, and something we are excited about. NAS for RL is a nascent field.

[Meta-Review · NeurIPS 2020]

The paper has been actively discussed after the rebuttal that the reviewers found useful and actionable (e.g., reply about the scalability with respect to larger population sizes). The paper is recommended for acceptance. All reviewers have acknowledged that the paper is making a step towards model-based online hyperparameter optimization and it is likely to inform future research. One suggestion of the reviewers is to possibly move the supervised learning evaluation to the appendix.